# A Retrospective Clinical Trial Regarding Oral Rehabilitation Diagnosis Strategies Based on Stomatognathic System Pathology

**DOI:** 10.3390/biomedicines11020622

**Published:** 2023-02-19

**Authors:** Iulian Costin Lupu, Laura Elisabeta Checherita, Magda Ecaterina Antohe, Ovidiu Stamatin, Silvia Teslaru, Tudor Hamburda, Eugenia Larisa Tarevici, Bogdan Petru Bulancea, Mioara Trandafirescu, Cristina Gena Dascalu, Magdalena Cuciureanu, Irina Gradinaru, Lucian Stefan Burlea, Alina Elena Jehac

**Affiliations:** 13rd Dental Medicine Department, Faculty of Dental Medicine, “Grigore T. Popa” University of Medicine and Pharmacy, 700115 Iasi, Romania; 22nd Dental Medicine Department, Faculty of Dental Medicine, “Grigore T. Popa” University of Medicine and Pharmacy, 700115 Iasi, Romania; 31st Department of Morpho-Functional Sciences, Faculty of General Medicine, “Grigore T. Popa” University of Medicine and Pharmacy, 16 Universității Street, 700115 Iasi, Romania; 4Department of Medical Informatics, “Grigore T. Popa” University of Medicine and Pharmacy, 16 Universității Street, 700115 Iasi, Romania; 5Department of Pharmacology, Faculty of Dental Medicine, “Grigore T. Popa” University of Medicine and Pharmacy, 16 Universității Street, 700115 Iasi, Romania

**Keywords:** stomatognathic system, diagnoses and examinations, rehabilitation outcome, periodontal diseases, mandibular diseases

## Abstract

Introduction: Orofacial pain is a common occurrence in daily dental practice; it is frequently attributed to temporomandibular dysfunction, one of its major causes, followed by pathology of the salivary glands, without avoiding interference at the level of the pain pathways caused by complications of periodontal pathology. The main objective of this study is to identify an important cause of pain in the oral–maxillofacial territory by quantifying the changes at the salivary glandular level using stereological methods. The secondary objective of the present research is to identify the implications of periodontal changes as a consequence of salivary quantitative and qualitative changes, quantified using periodontal indices, on the balance of the temporomandibular joint, dysfunction of it being an important cause of facial pain and having a profound impact on the complex oral rehabilitation algorithm of each clinical case, a condition evaluated with the analysis of the results of the Souleroy questionnaire. Material and methods: In a retrospective study, we evaluated the clinical results obtained after applying complex rehabilitation treatment to 35 subjects, 20 women and 15 men with salivary and TMJ dysfunctions, selected between 2020 and 2021 from the Clinic of Maxillofacial Surgery, Iasi. Results and discussion: The most common symptoms of temporomandibular disorders (TMDs) that were identified through the Souleroy questionnaire were pain and different types of damage to the masticatory muscles. The most significant changes in elders are reported in the case of serous cells, which reduced their percentage volume from 46.7% to 37.4%. Conclusion: As regards stereological analysis in conjunction with histological images, there were significant changes in diameters, perimeters, and longitudinal axes in the adult patients as opposed to the elderly patients, which were also influenced by the type of pathology at this level. The scores recorded on the diagnostic Souleroy scale indicated a large number of patients with low efficiency and maximum stress levels: 20.0% in level 1, 25.7% in level 2, and 25.7% in level 3.

## 1. Introduction

The diagnosis of pain in the oral and maxillofacial territory represents a real challenge for dental medical practice because the territory concerned is common and the forms of manifestation may have similar irradiation trajectories which can create difficulties in identifying different pathologies. Thus, pathologies of the temporomandibular joint and salivary glands are responsible for pain in the oral and maxillofacial area, with a high prevalence and a sometimes disabling impact on the quality of life of these patients. The dynamics of saliva quality and quantity, directly dependent on the status of the salivary glands, have a decisive influence on the intra-oral status through the effects of cleaning and self-cleaning often associated with different degrees of periodontal damage, the complications of which can influence the pathways of pain manifestation at the orofacial level [1].

The holistic integrative approach to pain in the oral–maxillofacial territory represents a relevant starting point for the conduct of the present study. It is known that the influence of general pathology at the oral–maxillofacial level creates specific channels of investigation and identifies pathognomonic signs for a specific disease, thus the association between temporomandibular joint dysfunction and salivary gland pathology in patients diagnosed with rheumatoid arthritis has been demonstrated. It is important to identify the particular situations in which a differential diagnosis between the two clinical entities is required. Thus, particular attention should be paid to the junction of these pathologies and the specific therapeutic management of the manifest forms of pain in the oral–maxillofacial territory influencing the different types of oral rehabilitation. The interconnecting aspects at the oral–maxillofacial level in terms of the cascading effect of temporomandibular pathology are reflected in a significant decrease in saliva associated with salivary changes. All these aspects plead for an integrative diagnosis to identify the specificity of pain in temporomandibular joint pathology compared to that triggered at the level of the salivary glands with the influence of periodontal status elements that individualize an entire vision of the diagnosis [2,3].

Temporomandibular disorders (TMDs), a group of chronic conditions, are considered the second most common musculoskeletal condition (after chronic low back pain) resulting in pain and disability. Pain-related TMD can affect an individual’s daily activities, psychosocial functioning, and quality of life. Patients with these symptoms often seek consultation with dentists for their TMD, especially for TMD-related pain [1].

Diagnostic criteria for TMD with simple, clear, reliable, and valid operational definitions for history, examination, and imaging procedures are needed to render physical diagnoses in both clinical and research settings.

TMD is an increasingly common diagnosis in the population with overt symptoms affecting the quality of life, and the prevalence of this condition in the adult population varies according to geographical location and the age range affected. In Saudi Arabia, the prevalence of TMD among students aged between 20 and 25 years was found to be 37%, while in Lebanon, the estimates were between 19% and 59.5% following questionnaire-based assessments, and in Poland, the prevalence of the condition was found to be 55% in urban areas [4,5,6]. In a meta-analysis-based study including 21 articles on the prevalence of TMD, estimates of the prevalence in the adult population ranged from 4% to 31%, which emphasizes the idea of raising the alarm about the importance of TMD both in its context and within the concept of oral rehabilitation. In the framework of the therapeutic algorithm of oral rehabilitation, a correct evaluation of the temporomandibular joint is a relevant starting point for therapeutic success, the criteria considered being in full agreement with the protocols existing at this time in contemporary medical practice, a particular aspect being the integrative approach of all affected parameters. Another important cause of the onset of pain in the oral–maxillofacial area belongs to pathologies of the salivary glands that affect about 20% of the adult population [7,8,9].

Salivary changes at the qualitative and quantitative level influence the oral status at the level of dental periodontal indices, the negative effects being found in patients with a low salivary volume that attracts cascading effects located at the dental periodontal level [10,11,12]. There are also some correlations between the morphological and functional changes in the salivary glands through aging, in the absence of other systemic pathologies, and the onset of the dysfunctional syndrome of the stomatognathic system, including TMJ (temporomandibular joint), is an accentuation of these changes but also an additional complication in patients who already experience TMD. This aspect can be considered a special supportive, interdisciplinary approach to the management of patients from the perspective of complex oral rehabilitation [13,14,15].

The anamnesis questionnaires have begun to be widely used as very important tools in diagnosis. These represent a method of investigation based on information provided by patients. These questionnaires are designed to provide important facts about the history of pain or a condition. The relevant starting points for choosing this type of questionnaire are represented by the existence of several types of pain at the level of the stomatognathic system, the maxillofacial space, characteristics such as irradiation of the essential elements in the sphere of differential diagnoses of the submaxillary glands and persistent pain at TMJ, and damage of the periodontal support tissue [16,17,18].

The information provided will help the achievement of a correct and individualized diagnosis. In the literature, the best-known pain questionnaires are the MPQ (McGill Pain Questionnaire) [19] and AVS (analogous visual scale). The way pain assessment is performed using these scales is verbal-descriptive. Coping questionnaires (adaptation) to pain (adjustment), defense mechanisms, attributions, stress, anxiety, and depression have the role of determining the level of influence that pain has on the mental state and behavior and then help nuance psychotherapeutic intervention. The qualities of the questionnaires are considered superior to the anamnesis (interview).

The originality of the study consists in bringing the stereological method as a predictor of pain pathology triggered by salivary gland damage and in integrating the evaluation of the salivary glands in the differential diagnosis of pain, which manifests itself with a similar character in the oral–maxillofacial territory. Interdisciplinary assessment of pain in the oral–maxillofacial territory is extremely important and forms the basis for appropriate therapeutic planning so that the results are maintained in the long term.

The working hypothesis on which this research is based is that pain in the oral–maxillofacial territory is a symptom encountered in different pathologies, the most common being associated with temporomandibular joint pathology and salivary gland pathology, the evaluation aspects of diagnostic elaboration and treatment-plan staging being carried out in the context of a cumulative factor.

## 2. Aim of the Study

The main objective of this study is to identify an important cause of pain in the oral–maxillofacial territory by quantifying the changes at the salivary glandular level using stereological methods.

The secondary objective of the present research is to identify the implications of periodontal changes as a consequence of salivary quantitative and qualitative changes, quantified using periodontal indices, on the balance of the temporomandibular joint, the dysfunction of it being an important cause of facial pain and having a profound impact on the complex oral rehabilitation algorithm of each clinical case, a condition evaluated with the analysis of the results of the Souleroy questionnaire.

## 3. Materials and Methods

The study group included 129 patients hospitalized at the Maxillofacial Surgery and Oncology Clinic between 2020 and 2021 and diagnosed with salivary dysfunction and TMJ disorders. From this batch, 118 eligible patients were selected, of which those who also completed the full oral-rehabilitation stages and not only the pain-diagnosis stages were 35 patients who also responded positively to the follow-up stages.

All patients had to fill in the associated questionnaire and, with consent, they were able to obtain a specific evaluation before the treatment based on the computerized program. (The present study was approved by the Ethics Committee of “Gr. Th. Popa” University of Medicine and Pharmacy Iasi TD 312/2008 and No. 176/17.04.2022. All protocols were in accordance with the provisions of the Declaration of Helsinki.) Our study focused on submaxillary salivary glands and had two main areas of approach.

The patients included in our study are those who had not undergone chronic treatments at different stages of life, a very important and limiting criterion. The inclusion criteria for the cases were: patients suffering from salivary lithiasis, and pain in the oral maxillary territory.

The exclusion criteria for the cases were: non-lithiasis patients and another pain type external to that of the stomatognathic system or oral–maxillofacial territory. We evaluated the clinical results obtained after applying complex rehabilitation treatment to 35 subjects, 20 women and 15 men, selected between 2020 and 2021 from St. Spiridon Hospital, Clinic of Maxillofacial Surgery and Oncology Hospital, Iasi, with salivary and TMJ dysfunctions. It is a retrospective study with limits according to the flowchart for patient selection (Figure 1).

The first part revealed histological aspects. We studied submaxillary salivary glands using 35 samples harvested from the bodies of both genders, 20 women and 15 men, and various ages without pathological damage. The cases were divided into two groups of age: adults (25–59 years) and elders (60–72 years). The pieces were processed using the paraffin-embedding technique and colored with the usual (H&E) and special methods (Masson’s trichrome, PAS, and Alcian blue), which revealed differentiated structural components. A qualitative examination was performed of sections to remove any pathological aspects and to select the representative sections without processing artifacts.

The second, microanatomic quantitative study was conducted using an interactive digital video software product applied to histological sections selected using standard measurement to assess the dimensions of secretory components.

Standard measurements facilitated the assessment of dimensions like diameter, perimeter, area, long and short axis, and the roundness of the secretory units separately for the serous, mucous, and small needles.

Quantifications were conducted on representative sections varying between 10×-80× objective, measuring, in each case, 20 secretory units of each type.

For the stereological examination, the report automatically submitted the stereological volume aspect percentage, verifying the significance of the obtained data. Quantification was carried out only in the glandular lobules, without going into the interlobular spaces. A Weibel parallel grid stereological scale was used in our study; a test surface corresponding to 540 points on the test grid with Weibel parallels was registered with the 90× objective, with a distance between two points of 19.39 μm (Figure 2).

Concerning the clinical assessment, particularly important issues are related to the identification of the Ramfjord index PDI (periodontal index) (at the level of 1.6, 2.1, 2.4, 3.6, 4.1, and 4.4 teeth), gingival bleeding index (GBI), Silness and Loe plaque index (PLI), clinical attachment loss index (CAL), and the Community Periodontal Index of Treatment Needs (CPITN), which offers an in-depth view of present periodontal damage (Figure 3).

To investigate the arsenal of diagnostic tools, we used the Souleroy questionnaire for revealing the most common symptom and quantifying the optimization obtained after treatment.

This is a diagnostic scale used to evaluate the treatment efficiency by placing the patient in a stage of the pathological situation or the degree of optimization within a performed treatment (Table 1).

The statistical analysis was performed in SPSS 27.0. The Kolmogorov–Smirnov test was used to check the distribution of the numerical variables. The numerical variables were expressed through average values and standard deviations, and the category values were expressed through absolute frequencies and percentages. The Pearson Chi-squared test was used to compare categorical variables within samples, and the t-Student test was used to compare numerical variables with normal distributions within the samples; Mann–Whitney and Wilcoxon Signed Ranks tests were used to compare numerical variables with non-normal distributions within the samples. The value of *p* < 0.05 was considered statistically significant and the value of *p* < 0.01 was considered highly statistically significant. The consent of all participants was obtained to meet the protocol of the scientific university ethical committee.

## 4. Results

The results of the study provide the data in their analytical sequence, starting with the changes identified by correlating the stereological analysis with the histological analysis of the salivary glands evaluated, followed by the data related to the periodontal indices analyzed according to salivary qualitative and quantitative changes as a natural consequence of salivary gland pathology. In the next step, the results relate to the analysis of the Souleroy questionnaires with profound implications for the identification of the pain factor at the temporomandibular joint level. All these elements are reflected in the way in which complex oral rehabilitation is influenced by the intra-oral status, the type of existing prosthesis, and the quality of the dental–periodontal support, as well as the way in which periodontal parameters are improved with specific antibiotic therapy, in the context of the integrative context of diagnosis and therapy.

The importance of microanatomical studies from a qualitative point of view is reflected in the detailed aspects of the structure of the salivary glands, a structure with deep implications for the diagnostic and therapeutic areas (Figure 4).

Structurally, the major salivary glands are made up of numerous lobes made up in turn of lobules and connected by dense connective tissue in which excretory channels and neurovascular elements are found. The structural differences between the three pairs of major salivary glands are documented in the literature, the parotid gland being a serous type, mixed submaxillary gland and having sublingual mucosa cell representation.

The presence of cellular condensations in the histological evaluations of the material studied, as well as acinar dimensional changes, serve as starting points and indications of inner functional deficiencies determined by functional changes.

The values of the standard measurements obtained were introduced in centralizing tables according to the parameter, while the results were checked statistically.

The most significant values were the area of the needles, the perimeter, the diameter, and the long and short axes according to which the roundness of the studied needles was assessed.

For submaxillary glands, a predominantly serous structure has been observed containing mucous cells as well.

In particular, serous acinus, rarely mixed acinus, serous cells forming the crescents, and rarely pure mucous acinus could be noticed. Striated ducts were noticed as being abundant.

Special stains reveal the existence of connective septa, which divide the parenchyma but are less obvious than in the parotid gland literature.

The old connective septa are thicker and branched. In addition, they are separated from fat which sometimes penetrates the lobules.

Stereology affords the appreciation of the percentage volumes of intralobular structures. The reference structures were established: serous and mucosal cells, interlobular excretory duct cells, blood vessels, and connective tissue. The statistical and stereological aspects of the components of the salivary gland structure are presented in Table 2.

The most significant changes in elders are reported in the case of serous cells, which reduce their percentage volume from 46.7% to 37.4%; connective tissue, which becomes more abundant (increases from 12.2% to 17.9%); and the appearance of adipose tissue (7.2%), the latter replacing the functionally specialized elements.

The diameters, perimeters, and longitudinal axes vary between the adult (25–59 years) and elderly patients (60–72 years) as follows (Table 3):

We identified a highly significant decrease in elders in the diameter, perimeter, and long axes of serous acini; the diameters of mucous and mixed acini are also diminished in elders, but not quite significantly, as well as the perimeter of mixed acini, while the long axes of mixed acini are also diminished, but statistically significantly; the perimeters and long axes of mucous acini are slightly increased in the elders compared with the adults. These variations related to age are justified by pathological and morphological changes.

For the evaluation of the damaged dental–periodontal status condition, we used systematization through mean values indices.

Regarding the intraoral clinical examination of the patient, the first important sign observed was periodontal damage, with a horizontal bone line at 33.10% and a vertical bone line at 27.93%.

We identified statistically significant differences between adults and elders for all periodontal indexes investigated, except the PLI index—generally, all these indexes increased in elder patients; the most obvious discrepancies were observed in the case of the PDI and GBI indexes (Table 4).

In summary, the elder patients in our study group displayed moderate to abundant levels of plaque compared to mild to moderate plaque in the adult patients; all elder subjects were diagnosed with moderate to severe plaque-induced gingivitis compared to mild to moderate plaque-induced gingivitis in the adult patients; most of the elder patients were diagnosed with moderate to severe chronic periodontitis compared to mild to moderate chronic periodontitis in the adult patients.

Inflammation is thus correlated to a decrease in the rate of salivary flow, with changes in the salivary glands that induce pathological aspects at the local, occlusal, and TMJ levels.

The results based on the statistical analysis of the data resulting from the questionnaires on pain symptomatology were corroborated with the qualitative and quantitative evaluations represented by the stereological and histological investigations and the quantification of the dental periodontal and loco-regional indices at the TMJ level.

Before proceeding to oral rehabilitation, we applied the Souleroy questionnaire as an initiator of the therapeutic plan, solving the first stage of mental rebalancing and craniomandibular cranial repositioning with the necessary therapeutic arsenal and then the morpho-functional aspect itself (Figure 5).

The main and common symptoms of TMDs very frequently consist of pain and accentuated stress. Related to this, the scores recorded on the diagnostic Souleroy scale indicate a large number of patients with low efficiency and maximum stress levels: 20.0% in level 1, 25.7% in level 2, and 25.7% in level 3. In the case of the adult patients, the situation is slightly better than in the case of the elders: only 1 adult patient has the minimum score of 1 (5.9%) compared with 6 elders (33.3%), 3 adults have a score of 2 (17.6%) compared with 6 elders (33.3%), and only 2 elders have an initial efficiency score of > 3 (11.2%) compared with 8 adults (47.0%). These differences are not statistically significant (Pearson Chi-squared *p* = 0.133).

Rehabilitation begins with pain relief and stress reduction, and then it continues with treatments to stabilize and reposition the structures as a curative principle in a complex oral rehabilitation plan. Concerning morpho-functional rehabilitation, we performed prosthetic rehabilitation followed by rehabilitation at the TMJ and muscular levels in the corresponding stomatognathic area, as depicted in Table 5.

In the context of restorative therapy, the biological tolerance of tissues is of particular importance, being a complex issue with many clinical implications. In the context of inappropriate restorative treatments, the biological tolerance of surrounding tissues could be followed by clinical implications, often leading to disruption of the existing oral biological balance. The emergence of local complications has led the literature in recent years to become increasingly concerned with the complex process involved in their integration into the biology of the oral cavity and the body. This biological aspect is imposed by the very different reactivity of tissues to the presence of any restorative treatment. The success of prosthetic therapy depends, on the one hand, on periodontal health and, on the other hand, on compliance with the principles of technological implementation and the choice of restorative material.

Periodontal pre-prosthetic training consisted of etiological treatment, followed by periodontal surgery to remove residual periodontal pockets in the patient group, associated with targeted systemic antibiotic therapy in the case of active pockets, according to the results of determining periodontopathogens with PCR tests. After that, we applied endodontic conservation techniques to prepare the dental structures in order to build the fixed and mobile prosthetic rehabilitation interventions. The fixed prosthesis treatment (34.3%) is definitely more reliable and comfortable for elderly patients with TMD-associated dysfunction and submaxillary gland pathology (38.9%). Partial mobile rehabilitation was suitable in 40.0% of the patients, again more often in elder patients (55.6%) than in the case of the adults (23.5%), and the total mobile treatment was recommended for 2 patients (5.7%), both elders (11.1%), being also combined with a special treatment for prosthetic field lubrication because of the dysfunctionality of the salivary products. The prosthetic treatment was also combined, when necessary, with endodontic treatment (in 60.0% of cases) or surgical interventions (in 48.6% of cases); all patients received balneo-physio-kinesiotherapy. The mixed rehabilitation was also accompanied, when necessary, by mouth guards, oral medication, and prosthetic treatment for temporomandibular disorders with an average level of tolerability and success.

The specific treatment procedure for inflammation of the salivary glands and pain is represented by antibiotic therapy (Table 6)—dicloxacillin (37.1% of cases), first-generation cephalosporins (40.0% of cases), clindamycin (22.9% of cases), or vancomycin (17.1% of cases)—which became necessary because of the increased prevalence of methicillin-resistant *Streptococcus aureus*, especially among elderly people living in asylums. The infection is due to a virus, such as herpes; the treatment is usually symptomatic but may include antiviral drugs (prescribed in our context in 20.0% of cases)

Regarding surgical interventions (28.6% of cases), abscesses required drainage; sometimes, a superficial parotidectomy or excision of the submandibular gland was indicated in patients with chronic or recurrent sialadenitis. In addition, because sialadenitis usually occurs after decreased saliva flow (hyposecretion), patients were usually advised to drink plenty of fluids and eat or drink things that trigger saliva flow. Warm compresses and gland massage can also be helpful if the flow is obstructed in some way.

The results regarding the diagnostic Souleroy scale, obtained after applying the rehabilitation procedures and the antibiotic therapy, are depicted in Figure 6.

We noticed a certain decrease in stress levels—the scores on the Souleroy scale vary between 5 and 10 after treatment compared with 1–7 before treatment. Most patients have very good scores (8—28.6% of cases or 9–10—48.6% of cases). This improvement was noticed for both age groups investigated, even if the results are better in the adult patients (64.7% with maximal scores of 9 or 10) compared with the elder patients (33.4% with maximal scores of 9 or 10). The differences between age groups are not statistically significant either in this case (Pearson Chi-squared *p* = 0.262), proving that the patients had a generally good response to the prescribed treatment.

We also performed a comparative analysis of the Souleroy scale score evolution by age group in relation to the treatment prescribed (rehabilitation procedures and antibiotic therapy) in order to evaluate the efficiency of specific interventions. The obtained results are described in Table 7.

The Souleroy scale scores significantly decreased after treatment in all cases, globally and separated by age group, some specific elements being also present. The best score improvements were observed in adult patients with fixed or partial prostheses, while the weakest improvement was observed in elder patients with mobile prostheses, with poor results also being noticed in the case of elder patients with partial prostheses. The endodontic treatment led to moderate improvements in the Souleroy scale score, as well as the surgery interventions—in this case, the improvements were even smaller, particularly for the adult patients. Among antibiotics, the best improvements in Souleroy scale scores were noticed in the adult patients who received dicloxacillin or vancomycin and the weakest in the adult patients who received antivirals. The elders’ response to antibiotics was weaker compared with the adults’ response; cephalosporine had a moderate effect on Souleroy scale scores in both age groups, as well as clindamycin; the antivirals registered a slight effect.

## 5. Discussion

In light of histological examinations performed on the salivary glands of healthy subjects, according to old literature from 1978 to 2000, Dayan revealed a significant reduction in the acinar volume and an increase in the volumes of stromal components, the highest volume being occupied by the inflammatory infiltrate [20]. The major changes consist of a reduction in the volume of acini and the concomitant increase in ductal volume, in contrast with expectations, with salivary secretion remaining unchanged or minimally altered. Experimental studies in rats have shown that although the volume of acini is reduced, cells can synthesize secretory proteins at a high level [21,22].

The analysis of the normal structure of the salivary glands enables a morphogenetic interpretation of the histological types of pathological variabilities in salivary glands.

It was observed that the stereological quantification performed shows differences in the secretory structural composition of the glands: in the submandibular, the percentage volume of serous cells predominates (42.0%).

All types of glandular acini (serous, mucous, or mixed) atrophy in the elderly, the most affected being serous acini. Their damage could explain the increase in the viscosity of saliva in the elderly and the reduction of its ability to moisturize and wash the oral cavity.

Standardized measurement of the acinar structures according to age and concomitant pathology helps us determine the structural changes that can influence a permissive and objective treatment in choosing the optimal rehabilitation therapeutic solution to facilitate the necessary functionality.

The percentage of connective tissue increases in the elderly compared to adults; the septa thicken and penetrate deep inside the lobe along with the excretory ducts, compressing the secretory units. In the elderly, the secretory structures are reduced in size, as is their percentage volume, being replaced by adipose tissue that dissociates not only the secretory elements but also the connective tissue. The system of excretory ducts becomes irregular in both salivary glands studied and increases its volume by the lumens [23,24]. As a result, salivary stasis occurs, and participation in the processes of active ion transport—the formation of definitive, secondary saliva—is altered [25,26].

Today, the evolution of methods for assessing salivary gland status has made remarkable progress, with digital subtraction sialography becoming an increasingly common paraclinical method for assessing salivary gland morphology and identifying specific pathology. The choice of salivary gland assessment methods is fully in line with the availability of specialist services and especially with the socioeconomic status of the patients assessed.

Periodontal diagnosis and control are based on clinical parameters to a large extent. The direct clinical diagnosis affects decisions to initiate treatment and the selection of methods and outlines the topographic area of application. Dentists also evaluate the outcome of treatment and attempt a long-term prognosis based on clinical parameters. Periodontal pocket formation and loss of attachment are pathognomonic signs of periodontal disease. Reduction in periodontal pocket depth and attachment gain are the goals of periodontal therapy. Periodontal pocket probing emerges as the obvious method for diagnosing disease and evaluating therapy. The main parameters assessed using periodontal probing are periodontal pocket depth (the distance between the marginal gingiva and the sulcus/periodontal pocket bottom), gingival recession (the distance between the enamel-cement junction and the marginal gingiva), and clinical attachment level (the distance between the enamel–cement junction and the sulcus/periodontal pocket bottom). The data obtained with digital methods provide useful information about the status of the salivary glands and their implications for the stomatognathic system.

The treatment of temporomandibular disorders should address the problems identified in the subjective complaints of patients, especially the acute ones and those in the inflammatory stage, with the aim of relieving pain first, as well as muscle spasms [27,28]. In the meantime, pain begins to decrease or disappear as influenced by various procedures, restoring jaw movement and alignment and craniomandibular repositioning [29,30]. The correction of posture is essential and should address the position of the head, neck, shoulders, and tongue [31,32,33]. The patient can perform exercises to improve jaw coordination, stability, and alignment [34,35]. We observed a significant percentage of rehabilitation in complex treatment involving people who had also had temporomandibular disorder problems and a significant response of 48.6% of patients having a good response to the therapeutic approach (Souleroy scale scores of 9 and 10); the rehabilitation of salivary dysfunction required antibiotics, dicloxacillin (37.1%) and cephalosporines (40.0%) being the most used treatments with substantial benefits noticed. Moreover, a very small value of salivary resting flow due to changes in salivary gland parenchymal attributable to microvascular glandular inflammation and edema was also noted, with patients having difficulties with chewing, swallowing, and speech.

Additionally, in order to make functionality more efficient and solve other associative and stressful causes, we can also establish an occlusion plan: the prevention or correction of the mandibular dysfunctions through prosthetic rehabilitation [36,37].

The approach in the literature to the assessment of salivary gland morphology as correlated with saliva type is reflected in the studies conducted by Chen et al., who used the combination of ultrasonography with virtual touch quantification for the diagnosis of Sjögren syndrome. The results of the study indicate that the assessment of salivary gland damage, predominantly in the submandibular and parotid glands, contributes significantly to the diagnosis of Sjögren syndrome. Therefore, the development of non-invasive and accurate diagnostic strategies for salivary gland morphology and pathology would contribute significantly to the early diagnosis of Sjögren syndrome associated with markedly decreased saliva quantity. In future studies, we aim to assess the parameters involved in triggering the pain factor at the temporomandibular joint of the salivary glands and the periodontal level using digital techniques fully quantifying the initial status of the therapeutic need and the therapeutic possibilities in the context of the initial variables. Furthermore, we intend to include in the statistical analysis predictability studies that provide the limits within which the initial parameters characterizing the temporomandibular joint, salivary glands, and periodontal pathology evolve, with an assessment of the degree of oral rehabilitation according to specific therapeutic methods and evaluation of long-term outcomes [38,39].

As humans get older, the submaxillary glands undergo similar quantitative changes, the decrease in salivary flow and the change in saliva composition being challenged by qualitative and quantitative structural changes. Because most of the successful therapies are dependent on parenchymal gland impairment, it is essential to know the structure to create a track to new therapeutic approaches. Computerized quantitative methods play a particularly important role in quantifying age-dependent changes in the parenchymal and stromal structure of the submandibular glands in selected subjects without systemic disease by digital diagnosis, with profound implications for our diagnostic management plan [40].

This outline circumscribes normal from pathological aspects and creates optimal premises and more precise approaches for the patients involved in the study, as well as in the case of overlapping associative pathologies of temporomandibular disorders, and generates a diagnostic panorama that can justify complex oral rehabilitation as well as its integrated and integral impact.

It is important to have an adequate comprehension of the diverse causes of salivary gland pathology to develop a systematic approach that includes collaboration with physicians to facilitate interdisciplinary patient care, which involves systemic conditions and medication.

Daily practice shows that the elements of therapeutic failure of oral rehabilitation are associated with the absence of an integrative assessment of the pain phenomenon caused by both temporomandibular joint pathology and salivary gland pathology, requiring a precise assessment and long-term monitoring. In order to prevent relapses, the evolving aspects of advanced techniques and technologies for the detection of oral–maxillofacial pathology have a definite impact on the accuracy of integrative pain diagnosis at this level. One aspect that should not be overlooked is that which is related to social cases and the low socioeconomic level of the patients. In these categories, the methods anchored in the register of stereology in conjunction with various classical imaging techniques at the level of the temporomandibular joint can lead to remarkable results in terms of prophylaxis, interceptive diagnosis, and complex long-term oral rehabilitation.

**The limitations of the study** concern some aspects related to the fact that the patients in the initial study group, once they had resolved the pain phenomenon, did not continue with the rehabilitation stages, thus creating gaps in the consistency of the results and the completion of the questionnaires related to pain perception.

## 6. Conclusions

We noticed in our study the existence of pathology in the salivary glands, expressed by the presence of pain easily overlapping with dysfunctional syndrome pain, as well as symptoms such as decreased salivary flow.

As regards stereological analysis in conjunction with histological images, there were significant changes in the diameters, perimeters, and longitudinal axes in the adult patients as opposed to the elderly patients, which were also influenced by the type of pathology at this level. The most significant changes in elders were reported in the case of serous cells, which reduced their percentage volume from 46.7% to 37.4%.

The scores recorded on the diagnostic Souleroy scale indicated a large number of patients with low efficiency and maximum stress levels: 20.0% in level 1, 25.7% in level 2, and 25.7% in level 3.

In our study, the Souleroy scale scores significantly decreased after treatment in all cases, globally and separated by age group, some specific elements being also present. The best score improvements were observed in the adult patients with fixed or partial prostheses, while the weakest improvement was noted in the elder patients with mobile prostheses, with poor results being also noticed in the case of the elder patients with partial prostheses.

The polyvalent approach to the pathology of the salivary glands is essential in correlation with changes at the salivary level. The methods of quantifying the acinar changes anchored in digital methods offer additional precision and manifest as an alarm signal in the integration of these changes in the case of stomatognathic system homeostasis.

## Figures and Tables

**Figure 1 biomedicines-11-00622-f001:**
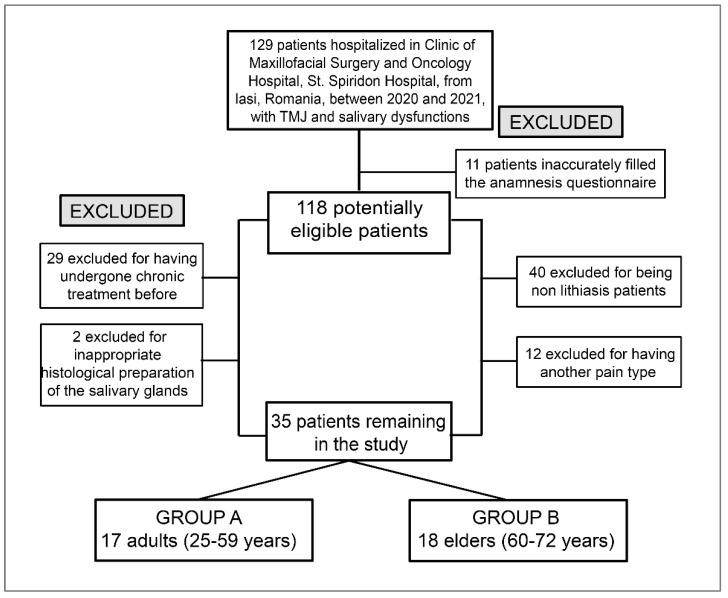
The flowchart for patient selection.

**Figure 2 biomedicines-11-00622-f002:**
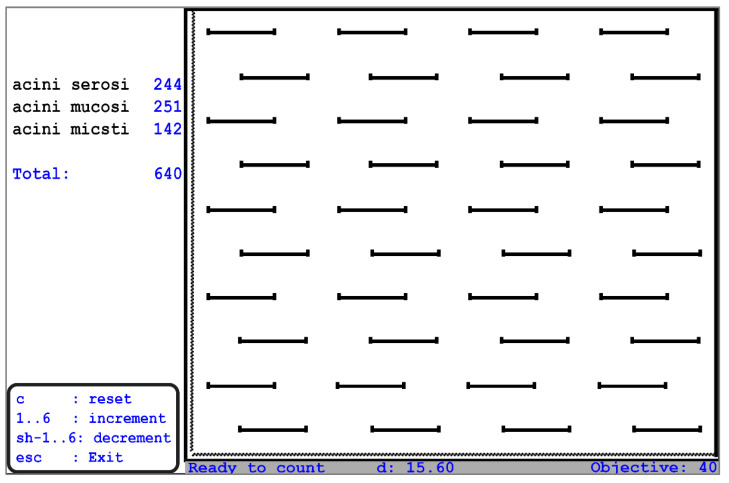
Computerized graded scale—a stereological tool for a computerized assessment of the percentage volumes of Weibel grid parallel—intralobular structures (“Grigore T. Popa” University of Medicine and Pharmacy, Iasi Anatomy Department).

**Figure 3 biomedicines-11-00622-f003:**
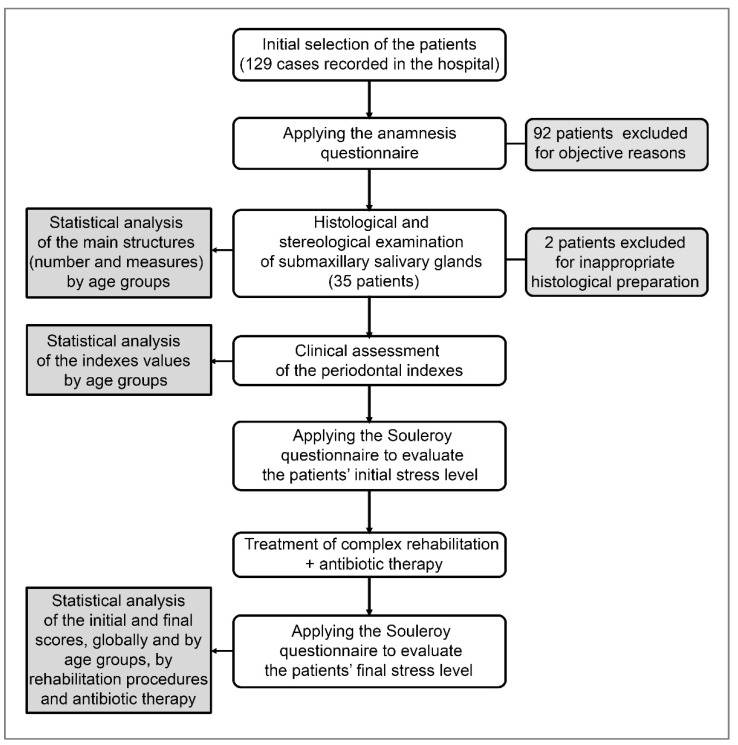
The study workflow.

**Figure 4 biomedicines-11-00622-f004:**
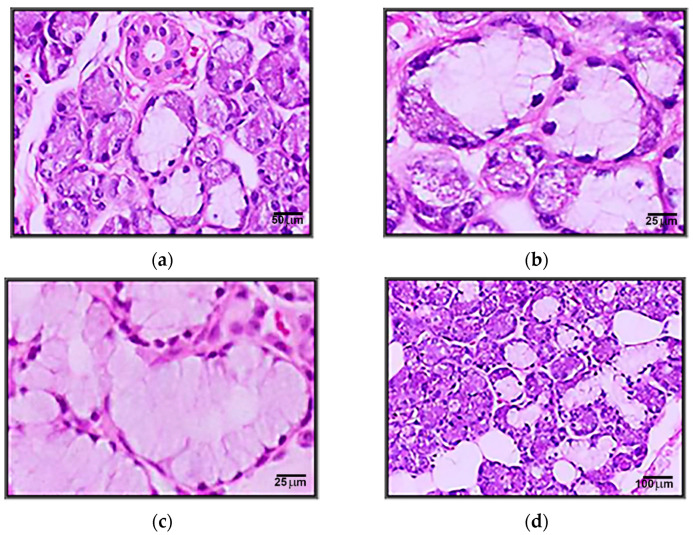
(**a**) Microscopic appearance of the submandibular gland with the presence of serous and mixed acini (200xH&E). (**b**) Coexistence of serous and mucous portions in the same acini (crescent) in the sub-mandibular gland (400xH&E). (**c**) Microscopic appearance of an area with mucous acini in the submandibular gland (400xH&E). (**d**) Microscopic appearance of the submandibular gland in the elderly with an increasing amount of adipose tissue (100xH&E).

**Figure 5 biomedicines-11-00622-f005:**
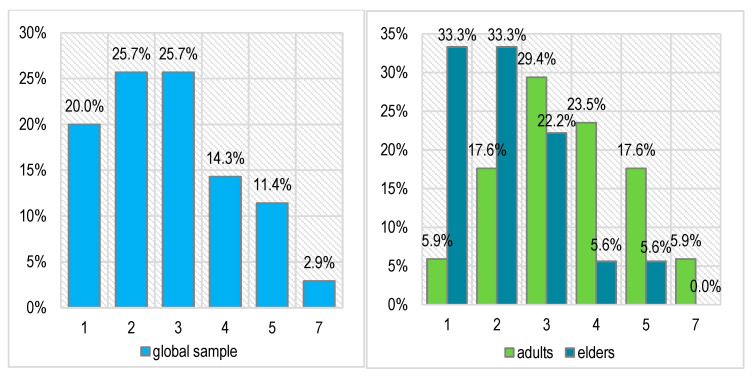
The patients’ initial stress level on Souleroy scale (globally and separated by age group; 0—no efficiency to 10—maximum efficiency).

**Figure 6 biomedicines-11-00622-f006:**
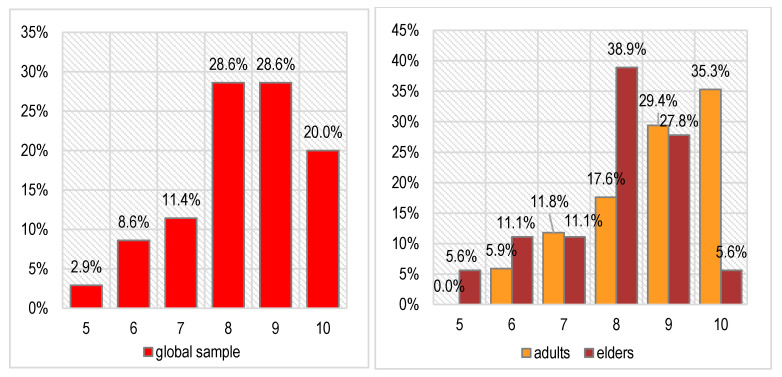
The patients’ final stress level on Souleroy scale (globally and separated by age group; 0—no efficiency to 10—maximum efficiency).

**Table 1 biomedicines-11-00622-t001:** The diagnostic investigative questionnaire tool—Souleroy scale (diagnostic scale in order to place the patient in a stage of the respective pathological situation or the degree of optimization within a performed treatment).

Efficiency on spasm	Good	2
	Medium	1
	None	0
Efficiency on muscle tone	Good	2
	Medium	1
	None	0
Length of efficiency	>3 days	2
	1–3 days	1
	<1 day	0
Side effects	No	1
	Yes	0
Evolution (after several coats)	Improvement	1
	Aggravation or no improvement	0
Pain	Complete removal	2
	Partial removal or improvement	1
	Lack of efficiency	0

**Table 2 biomedicines-11-00622-t002:** Stereological ratio of the percentage volumes of the main structures in a submandibular gland in the analyzed sample (globally and separated by age group).

	Total		Adults		Elders	
	n	%	n	%	n	%
Hits serous cell	227	42.0%	126	46.7%	101	37.4%
Hits mucous cell	118	21.9%	62	23.0%	56	20.7%
Hits duct cell	85	15.7%	44	16.3%	41	15.3%
Hits connective tissue	81	15.0%	33	12.2%	48	17.9%
Hits vessels	9	1.7%	5	1.9%	4	1.5%
Hits adipose tissue	20	3.7%	0	0.0%	20	7.2%
Hits nerve	0	0.0%	0	0.0%	0	0.0%
Total	540	100.0%	270	100.0%	270	100.0%
	Chi-squared test:		*p* < 0.001, highly statistically significant			

**Table 3 biomedicines-11-00622-t003:** Dimensions of quantified secretory units in the submandibular glands.

TYPE	Diameter		Perimeter		Long Axes	
	Adults M ± SD	Elders M ± SD	Adults M ± SD	Elders M ± SD	Adults M ± SD	Elders M ± SD
Serous acini (n = 227) Mucous acini (n = 118) Mixed acini (n = 195)	57.58 ± 8.50	51.33 ± 7.50	207.04 ± 29.46	172.80 ± 26.46	82.39 ± 13.71	62.26 ± 11.22
	*p* < 0.001 **		*p* < 0.001 **		*p* < 0.001 **	
	79.82 ± 8.92	77.29 ± 15.22	275.83 ± 35.88	281.35 ± 65.61	105.29 ±18.17	111.48 ± 25.36
	*p* = 0.267		*p* = 0.567		*p* = 0.128	
	65.32 ± 10.94	63.88 ± 11.62	226.99 ± 40.80	221.18 ± 40.22	87.96 ± 19.36	82.92 ± 15.52
	*p* = 0.383		*p* = 0.324		*p* = 0.045 *	
	*t*-Student test for independent samples: * *p* < 0.05 statistically significant; ** *p* < 0.01 highly statistically significant					

**Table 4 biomedicines-11-00622-t004:** The values of periodontal indexes recorded during clinical examination (globally and separated by age group).

	PDI Index					*p*-Value
		1: n (%)	2: n (%)	3: n (%)	4: n (%)	
Total (n = 35)		5 (14.3)	12 (34.3)	12 (34.3)	6 (17.1)	0.007 *
Adults (n = 17)		4 (23.5)	9 (52.9)	4 (23.5)	0 (0.0)	
Elders (n = 18)		1 (5.6)	3 (16.7)	8 (44.4)	6 (33.3)	
	**GBI index**					
	0: n (%)	1: n (%)	2: n (%)			
Total (n = 35)	8 (22.9)	18 (51.4)	9 (25.7)			0.003 *
Adults (n = 17)	6 (35.3)	11 (64.7)	0 (0.0)			
Elders (n = 18)	2 (11.1)	7 (38.9)	9 (50.0)			
	**PLI index**					
	0: n (%)	1: n (%)	2: n (%)	3: n (%)		
Total (n = 35)	2 (5.7)	15 (42.9)	14 (40.0)	4 (11.4)		0.286
Adults (n = 17)	1 (5.9)	10 (58.8)	5 (29.4)	1 (5.9)		
Elders (n = 18)	1 (5.6)	5 (27.8)	9 (50.0)	3 (16.7)		
	**CAL index**					
		1: n (%)	2: n (%)	3: n (%)	4: n (%)	
Total (n = 35)		2 (5.7)	17 (48.6)	13 (37.1)	3 (8.6)	0.039 *
Adults (n = 17)		2 (11.8)	11 (64.7)	4 (23.5)	0 (0.0)	
Elders (n = 18)		0 (0.0)	6 (33.3)	9 (50.0)	3 (16.7)	
	**CPITN index**					
		1: n (%)	2: n (%)	3: n (%)		
Total (n = 35)		9 (25.7)	22 (62.9)	4 (11.4)		0.018 *
Adults (n = 17)		8 (47.1)	8 (47.1)	1 (5.9)		
Elders (n = 18)		1 (5.6)	14 (77.8)	3 (16.7)		
		Chi-squared test; * *p* < 0.05 statistically significant				

**Table 5 biomedicines-11-00622-t005:** The performed rehabilitation procedures (globally and separated by age group).

	Total		Adults		Elders	
	n	%	n	%	n	%
Fixed prosthesis	12	34.3%	5	29.4%	7	38.9%
Partial prosthesis	14	40.0%	4	23.5%	10	55.6%
Mobile prosthesis	2	5.7%	0	0.0%	2	11.1%
Endodontic treatment	21	60.0%	9	52.9%	12	66.7%
Surgery interventions	10	28.6%	4	23.5%	6	33.3%
Balneo-physio-kinesiotherapeutical interventions	35	100.0%	17	100.0%	18	100.0%

**Table 6 biomedicines-11-00622-t006:** The prescribed antibiotic therapy (globally and separated by age group).

	Total		Adults		Elders	
	n	%	n	%	n	%
Dicloxacillin	13	37.1%	7	41.2%	6	33.3%
Cephalosporines (1st generation)	14	40.0%	6	35.3%	8	44.4%
Clindamycin	8	22.9%	6	35.3%	2	11.1%
Vancomycin	6	17.1%	1	5.9%	5	27.8%
Antivirals	7	20.0%	2	11.8%	5	27.8%

**Table 7 biomedicines-11-00622-t007:** The comparative values of Souleroy scale scores by age group, rehabilitation procedures, and antibiotic therapy.

Stress Level on Souleroy Scale (0 No Efficiency–10 Maximum Efficiency)	Total (M ± SD)		Adults (M ± SD)		Elders (M ± SD)	
	Initial Score	Final Score	Initial Score	Final Score	Initial Score	Final Score
**Global score**	2.83 ± 1.47	8.31 ± 1.32	3.53 ± 1.46	8.76 ± 1.25	2.17 ± 1.15	7.89 ± 1.28
	*p* < 0.001 **		*p* < 0.001 **		*p* < 0.001 **	
**Rehabilitation procedures**						
Fixed prosthesis	2.92 ± 1.31	9.25 ± 0.087	3.60 ± 0.894	10.00	2.43 ± 1.40	8.71 ± 0.76
	*p* = 0.002 **		*p* = 0.039 *		*p* = 0.017 *	
Partial prosthesis	2.50 ± 1.23	8.00 ± 1.36	3.50 ± 1.29	9.25 ± 0.50	2.10 ± 0.99	7.50 ± 1.27
	*p* < 0.001 **		*p* = 0.066		*p* = 0.005 **	
Mobile prosthesis	1.00	7.00 ± 1.41	-	-	1.00	7.00 ± 1.41
	*p* = 0.180		-		*p* = 0.180	
Endodontic treatment	2.67 ± 1.39	8.43 ± 0.87	3.44 ± 1.67	8.78 ± 0.97	2.08 ± 0.79	8.17 ± 0.72
	*p* < 0.001 **		*p* = 0.007 **		*p* = 0.002 **	
Surgery interventions	2.80 ± 1.75	7.40 ± 1.65	3.50 ± 1.73	7.50 ± 1.29	2.33 ± 1.75	7.33 ± 1.97
	*p* = 0.005 **		*p* = 0.066		*p* = 0.027 *	
Balneo-physio-kinesiotherapeutical interventions	2.83 ± 1.47	8.31 ± 1.32	3.53 ± 1.46	8.76 ± 1.25	2.17 ± 1.15	7.89 ± 1.28
	*p* < 0.001 **		*p* < 0.001 **		*p* < 0.001 **	
**Antibiotic therapy**						
Dicloxacillin	2.92 ± 1.44	8.69 ± 1.44	3.57 ± 1.13	9.43 ± 0.54	2.17 ± 1.47	7.83 ± 1.72
	*p* = 0.001 **		*p* = 0.017 *		*p* = 0.027 *	
Cephalosporines (1st generation)	3.00 ± 1.62	8.21 ± 1.12	4.00 ± 1.67	8.33 ± 1.03	2.25 ± 1.17	8.13 ± 1.25
	*p* < 0.001 **		*p* = 0.027 *		*p* = 0.011 *	
Clindamycin	2.75 ±1.58	8.63 ± 1.51	3.17 ± 1.60	8.67 ± 1.75	1.50 ± 0.71	8.50 ± 0.71
	*p* = 0.010*		*p* = 0.026 *		*p* = 0.157	
Vancomycin	2.33 ± 0.82	8.00 ± 1.41	3.00	10.00	2.20 ± 0.84	7.60 ± 1.14
	*p* = 0.026*		-		*p* = 0.039 *	
Antivirals	2.00 ± 1.41	7.29 ± 1.38	1.50 ± 0.71	6.50 ± 0.71	2.20 ± 1.64	7.60 ± 1.52
	*p* = 0.017*		*p* = 0.157		*p* = 0.041 *	
	Wilcoxon Signed Ranks test: * *p* < 0.05 statistically significant; ** *p* < 0.01 highly statistically significant					

## Data Availability

Not applicable. All data presented in the article are available from the authors and have not been published.

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
