# Peer review of "A Retrospective Clinical Trial Regarding Oral Rehabilitation Diagnosis Strategies Based on Stomatognathic System Pathology"

_biomedicines, 2023, doi:10.3390/biomedicines11020622_

Round 1

Reviewer 1 Report

Thank you for the opportunity to read the manuscript "Integrated Strategies in Order to Establish Oral Rehabilitation

Diagnosis Status Based on Stomatognathic System Pathology" before publication. Below are my comments, which I hope will be helpful to the authors.

Title

- The title is too complex in my opinion. Please try to make it simpler and clearer.

- What is missing from the title is that the article reports a retrospective clinical trial. I recommend adding such information in the form chosen by the Authors.

Abstract

- Bolds in the abstract are inconsistent - please correct or remove them.

- Correct keywords according to MeSH.

Introduction

- You mention TMJs and the salivary glands as the main sources of pain in the maxillofacial area. Verify this fragment against sources and provide figures for the population from which you selected the study group. (In the next paragraph, you relate TMD to musculoskeletal disorders - this is an independent comparison, which I have no objections to.)

- You write about the need to establish clear TMD diagnostic criteria, but you do not refer to the existing ones. Take for example DC/TMD 2014 and if you think they are flawed, justify why you think so.

- You continue to write about the composition of saliva, salivary glands, periodontium, but the text is not consistent. I propose the first paragraph of the introduction about maxillofacial pain, the second about its potential causes, and the third about the available diagnostic methods with an indication of the one described by the authors.

- Currently, some of the introductory paragraphs are one-sentence, which should be avoided.

Aim

- The purpose of the study should be simple to understand and concise. Write it in one sentence so that the reader knows what you are researching and why after the first reading. If you want to keep two research goals, then define them as primary objective and secondary objective.

- Consider whether you are trying to force all your assumptions, research, results and conclusions from the entire project into one article. This approach is not attractive to the reader. The reader expects an article with a clear title, a clear purpose of research and a simply formulated conclusion. Try to reject redundant content throughout the manuscript. It may turn out that in fact your project dealt with several aspects, you achieved several goals and it is best to publish your results in a few concise articles.

Materials and methods

- The beginning of this section is a reflection on questionnaires. They should be moved to the introduction, possibly to the discussion. In the methodological section, there is no room for background presentation or discussion with the literature. Please correct this issue.

Results

- Adapt the presentation of the content of the entire manuscript to one of the established protocols, such as STROBE.

Discussion

- Limitations subsection is missing.

Conclusions

- Please move the content of this section to the discussion, leaving only 1-3 sentences of a precise answer to the research question (research objective) in the conclusions.

Back Matter

- Delete the duplicate contribution section.

- Specify the availability of data - write whether all of them are included in the body of the article, whether they are available from the authors or published elsewhere.

References

- Resign from items older than the last 5 years (unless absolutely necessary, e.g. still valid guidelines).

Author Response

Good afternoon, Thank you for taking the time to improve our exposing research, and we hope to follow your advice as it progresses.

Title

- The title is too complex in my opinion. Please try to make it simpler and clearer.-modified

 A Retrospective Clinical Trial Regarding Oral Rehabilitation Diagnosis Strategies Based on Stomatognathic System patology

- What is missing from the title is that the article reports a retrospective clinical trial. I recommend adding such information in the form chosen by the Authors.- modified  

Abstract

- Bolds in the abstract are inconsistent - please correct or remove them. -Done

  • - Correct keywords according to MeSH. -done

stomatognathic system; diagnoses and examinations; rehabilitation outcome; periodontal diseases; mandibular disease

Introduction

- You mention TMJs and the salivary glands as the main sources of pain in the maxillofacial area. Verify this fragment against sources and provide figures for the population from which you selected the study group. (In the next paragraph, you relate TMD to musculoskeletal disorders - this is an independent comparison, which I have no objections to.)

- You write about the need to establish clear TMD diagnostic criteria, but you do not refer to the existing ones. Take for example DC/TMD 2014 and if you think they are flawed, justify why you think so.-

- You continue to write about the composition of saliva, salivary glands, periodontium, but the text is not consistent. I propose the first paragraph of the introduction about maxillofacial pain, the second about its potential causes, and the third about the available diagnostic methods with an indication of the one described by the authors. – modified accordance with the recommendation, deletion of senseless pharagraphs indicatede  and adding new notion regarding the prevalence of TMJ.

- Currently, some of the introductory paragraphs are one-sentence, which should be avoided-.modified

Aim- The purpose of the study should be simple to understand and concise. Write it in one sentence so that the reader knows what you are researching and why after the first reading. If you want to keep two research goals, then define them as primary objective and secondary objective.

- Consider whether you are trying to force all your assumptions, research, results and conclusions from the entire project into one article. This approach is not attractive to the reader. The reader expects an article with a clear title, a clear purpose of research and a simply formulated conclusion. Try to reject redundant content throughout the manuscript. It may turn out that in fact your project dealt with several aspects, you achieved several goals and it is best to publish your results in a few concise articles.- Restructured  having a central objective and secondary one .

Materials and methods

- The beginning of this section is a reflection on questionnaires. They should be moved to the introduction, possibly to the discussion. In the methodological section, there is no room for background presentation or discussion with the literature. Please correct this issue. - Corrected

Results

- Adapt the presentation of the content of the entire manuscript to one of the established protocols, such as STROBE. -Done

Discussion

- Limitations subsection is missing. -Added

Conclusions

- Please move the content of this section to the discussion, leaving only 1-3 sentences of a precise answer to the research question (research objective) in the conclusions.-done

Back Matter

- Delete the duplicate contribution section.-done

- Specify the availability of data - write whether all of them are included in the body of the article, whether they are available from the authors or published elsewhere- done 

References

- Resign from items older than the last 5 years (unless absolutely necessary, e.g. still valid guidelines). -Intervention according to changes from  the structure of the actual  improuved  article .

Reviewer 2 Report

Integrated Strategies in Order to Establish Oral Rehabilitation Diagnosis Status Based on Stomatognathic System Pathology

  • The working hypothesis on which this research is based is that pain in the oro-maxillofacial territory is a symptom encountered in different pathologies, the most common  being associated with temporomandibular joint pathology and salivary gland pathology,  the evaluation aspects of diagnostic elaboration and treatment plan staging being carried  out in the context of a cumulative factor.
  • The topic is original , The style and quality of the work is very confusing
  • The main problem is not having a clear objective.

The present study aims the digital  stereo lithographic technique can be the basis for an accurate diagnosis in the periodontal  area while creating the correlative premises between the type of saliva and periodontal damage. We also aim to quantify the implications that can be found at the level of tem poromandibular joint functionality taking into account that any type of prosthetic reha-  bilitation interferes with the anatomoanatomic-functional particularities of each patient

  • Material and methods are confusing
    • Indicates disorganized procedures.
  • Inclusion and exclusion criteria are unclear. After applying complex rehabilitation treatment?????
  • Does not follow any guidelines
  • No control group
  • We studied submaxillary salivary glands    being associated with temporomandibular joint pathology ¿?????
  • Results  Should describe a general sample

  • The conclusions of the study are randomized, because the objective is not clear.
  • References OK

Author Response

                     Good afternoon,

Thank you for taking the time to improve our exposing research, and we hope to follow your advice as it progresses.

       The working hypothesis on which this research is based is that pain in the oro-maxillofacial territory is a symptom encountered in different pathologies, the most common  being associated with temporomandibular joint pathology and salivary gland pathology,  the evaluation aspects of diagnostic elaboration and treatment plan staging being carried  out in the context of a cumulative factor.

  • The topic is original ,
  • The style and quality of the work is very confusing  -are changed   and  restructured .
  • The main problem is not having a clear objective. - Added  specifich objectives

The present study aims the digital  stereo- lithographic technique can be the basis for an accurate diagnosis in the periodontal  area while creating the correlative premises between the type of saliva and periodontal damage. We also aim to quantify the implications that can be found at the level of tem poromandibular joint functionality taking into account that any type of prosthetic rehabilitation interferes with the anatomo-functional particularities of each patient.

  • Material and methods are confusing - restructured
    • Indicates disorganized procedures.  -reorganized

  • Inclusion and exclusion criteria are unclear. After applying complex rehabilitation treatment????? --- Rewrite the methodology section and  clearifing the criteria section .-
  • Does not follow any guidelines --  modified after guiding  directives
  • No control group – It was not our purpose to design our study as a case-control one. We dealed with patients with salivary and TMJ dysfunctions, for which it was mandatory to apply complex rehabilitaton treatment – and our purpose was to compare the adults and the elders in order to emphasize possible  differences between age groups, for the same pathology, as well as their response to treatment (complex rehabilitation).
  • It was not ethically to keep a sample of patients without treatment, in order to use them as control group. -Done

Round 2

Reviewer 1 Report

The authors addressed most of my suggestions and provided a revised version of the manuscript.

Author Response

Good afternoon. thank you very much for your appreciation and support.

Reviewer 2 Report

The changes have improved. It is a very confusing paper.

  • The style and quality of the work is very confusing The opinion that an article has serious flaws, make sure you read the whole paper
  • NO Highlights gaps in current understanding or conflicts in current knowledge
  • NO Establishes the originality of the research aims by demonstrating the need for investigations in the topic area
  • NO Gives a clear idea of the target readership, why the research was carried out and the novelty and topicality of the manuscript

Author Response

Good afternoon, thank you for your suggestions, we want to answer your recommendations as better as it possible :

The changes have improved. It is a very confusing paper.

  • The style and quality of the work is very confusing .The opinion that an article has serious flaws, make sure you read the whole paper.----------.we modified the structure  in  abstract ,introduction, results, discution and  conclusion .
  • NO Highlights gaps in current understanding or conflicts in current knowledge.

We debated at the discussion level some representative opinions found in the literature as references taken by us only as a main idea, our link being related to differential diagnosis in oro -maxillo -facial zone deficiencies.

  • NO Establishes the originality of the research aims by demonstrating the need for investigations in the topic area.

We want to have adjuvant methods of  identification patology approach ,this situation create links that can more easily identify interpretative guidelines in predictor support  and help practitioners find more accurate diagnoses in the oro- maxillo- facial area.

  • NO Gives a clear idea of the target readership, why the research was carried out and the novelty and topicality of the manuscripts.

A  mechanistic  of correlative approach that can have a prophylactic impact in the diagnoses, to existing complex clinical situations in this level .